# Watch the Mime Carefully! A Refractory Interstitial Lung Disease

**DOI:** 10.3390/diagnostics12071743

**Published:** 2022-07-19

**Authors:** Paolo Graziano, Paolo Fuso, Cristiano Carbonelli

**Affiliations:** 1Pathology Unit, Department of Services, Fondazione IRCCS Ospedale Casa Sollievo Della Sofferenza, 71013 San Giovanni Rotondo, Italy; paologratz@gmail.com; 2Department of Medical and Surgical Sciences, Institute of Respiratory Diseases, Policlinico Universitario “Riuniti” di Foggia, University of Foggia, 71121 Foggia, Italy; paolo.fuso91@gmail.com; 3Pneumology Unit, Department of Medical Sciences, Fondazione IRCCS Ospedale Casa Sollievo Della Sofferenza, 71013 San Giovanni Rotondo, Italy

**Keywords:** chest CT, interventional pulmonology, lung cancer diagnosis, interstitial lung disease, pulmonary epithelioid hemangioendothelioma, pulmonary pathology

## Abstract

Epithelioid hemangioendothelioma (EHE) is a rare neoplasm of a vascular origin which can arise in different locations such as the lungs, liver, soft tissue, and rarely, in the bones. In the lungs, pulmonary hemangioendothelioma (PEH) shows a variable clinical behavior, displaying a range from either an asymptomatic course to a highly aggressive progression with metastases. Based on radiological features, PEH differential diagnosis mainly includes primary or metastatic lymphangitic carcinomatosis, granulomatous infections, and diffuse interstitial lung diseases where ground glass pattern predominates. In this case, a transbronchial biopsy and subsequent histological and immunohistochemical analysis allowed for the attribution of the scenario to a pulmonary epithelioid hemangioendothelioma. Clinicians should always consider bronchoscopy as a useful and effective tool to better investigate indeterminate and questionable clinical pictures, sparing patients the morbidity and mortality associated with more invasive techniques such as surgical or thoracoscopic biopsy.

**Figure 1 diagnostics-12-01743-f001:**
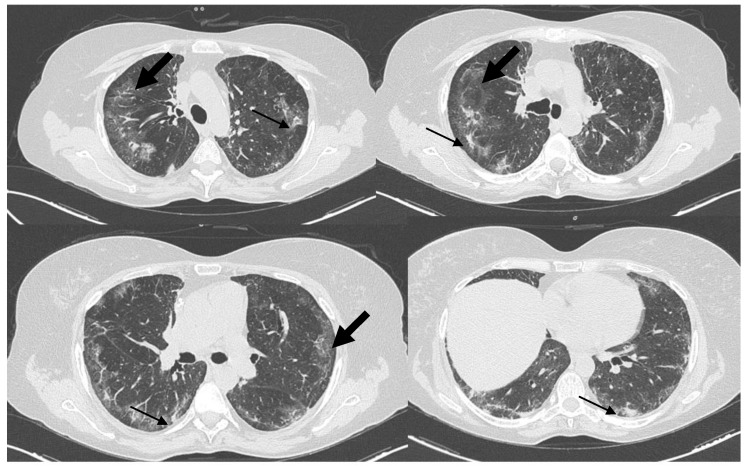
High resolution chest CT (HRCT), on lung window settings, showing an interstitial lung pattern displaying multilobar ground-glass opacities (large arrows); consolidations with traction bronchiectasis (small arrows) in both lungs, with peripheral and posterior distributions; subpleural sparing; and absence of lymphadenopathy for a 50-year-old nonsmoker woman, who was admitted to a general medicine ward of a peripheral hospital, on December 2020, for a 1-month history of increasing dyspnea and dry cough. In her medical history, the patient was treated in 2016 by radiotherapy and chemotherapy for a Non-Hodgkin’s Lymphoma (NHL), which was currently in follow up, and had hypertensive cardiopathy and dyslipidemia, both of which were under drug treatments. The laboratory tests were non-specific: there was a modest increase in white blood cells and CRP, and multiple SARS-CoV-2 PCR tests from nasopharyngeal swabs tested negative. A physical exam revealed a shortness of breath with a 93% oxygen saturation in the ambient air, a respiratory rate of 25 breaths per minute, a heart rate of 85 beats per minute, a blood pressure of 140/80 mmHg, and apyrexia. On an auscultation, fine bilateral and basal teleinspiratory crackles of the lung were revealed. A chest x-ray showed—mainly as a submantellar pattern—bilateral interstitial lung involvement plus the elevation of the right emidiaphragm. Thus, in the suspicion of a nonspecific interstitial pneumonia (NSIP) or an organizing pneumonia (OP), the patient began a long course of therapy with a high dose of corticosteroids, without any improvement. After almost four months, the patient was admitted to the Accident and Emergency Department of the Scientific Institute for Research and Health Care, ‘Casa Sollievo della Sofferenza’, for a new episode of worsening dyspnea and the presence of type 1 respiratory failure, which had been treated with oxygen therapy. Thus, she was admitted to the Department of Medical Sciences, Pneumology Unit, and a plethysmography highlighted a moderately severe restrictive deficit (TLC 56%, FVC 53%, FEV1 60%, IT 0.90) and a moderate reduction of DLCO (48%), and in the following days, she underwent to a second high resolution chest CT that showed an overall worsening of the diffuse interstitial involvement. Upon a contrast-enhanced full body CT scan, multiorgan disease was excluded. A bronchoscopy with a bronchoalveolar lavage (BAL) and transbronchial lung biopsies (TBB) from the middle lobe and the right lower lobes were performed. Upon the BAL, numerous macrophages but no microorganisms were detected. Epithelioid hemangioendothelioma (EHE) is a rare neoplasm of vascular origin which can arise in different locations such as the lungs, liver, soft tissue, and rarely, in the bones [1]. It is often incidentally detected, and its estimated prevalence is less than 1 in 1 million [2]. Pulmonary hemangioendothelioma (PEH) was first described in 1975 by Dail and Liebow and was originally named intravascular bronchioloalveolar tumor (IVBAT) [3]. The tumor shows a variable clinical behavior, ranging from an asymptomatic course to a highly aggressive progression with metastases. PEH usually affects young/middle-aged people, and its incidence is two times higher in women than in men and its prognosis is unpredictable [3]. Patients can remain stable for a long time or experience rapid disease progression until exitus, and the life expectancy may range from 1 to 30 years [4]. PEH is mainly incidentally discovered in young women, and some patients may present symptoms such as chest pain, pleuritic pain, cough, dyspnea, or rarely hemoptysis [5]. Typically, PEH occurs as bilateral multiple small solid and subsolid nodules in the lungs or pleura but can also present as multiple pulmonary reticulonodular lesions with interlobular septa inspissation, ground-glass opacities, or diffuse infiltrative pleural thickening [6].

**Figure 2 diagnostics-12-01743-f002:**
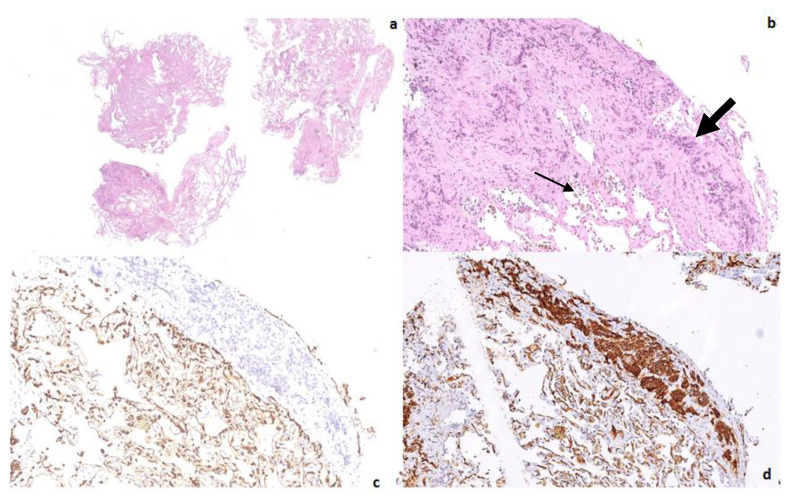
Transbronchial lung biopsy specimen revealed fragments of lung parenchyma characterized by subacute and fibrosing interstitial lung damage associated with chronic hemorrhagic lung injury (small arrow). Low and high-power magnifications demonstrated an interstitial and endovascular infiltration (large arrow) of epithelioid shaped cells, respectively, ((**a**,**b**) hematoxylin-eosin X2 and X13). The epithelioid cells resulted negative (**c**) for pancytokeratin (clone AE1-3 X13), and were strongly immunoreactive (**d**) for CD31 (X 12,5). Pulmonary hemangioendothelioma (PEH) was diagnosed; the patient was referred to a specialized Oncology Unit but died 10 months later. In almost all PEHs, a characteristic t(1;3) (p36; q23–25) translocation leading to a WWTR1-CAMTA1 fusion can be demonstrated [7]. Where a single, resectable pulmonary nodule is detected, a surgical approach could be an effective therapeutic option, whereas chemotherapy might be adopted in cases of multiple unresectable lung lesions [4]. Most of the previously published reports were diagnosed using open-lung or thoracoscopic specimens with confirmatory immunohistochemical stains of irregularly epithelioid-shaped cells, which were immunoreactive for CD31, ERG, and FLi-1 and negative for CD34, pancytokeratins, CK7, p63, chromogranin, synaptophysin, MUM1/IRF4, and TTF-1 [8], but a diagnosis has also been obtained by transbronchial samples [9]. Owing to the low incidence of PEH, there are no definite treatment guidelines. In fact, there is no single effective treatment, though spontaneous regression and a response to chemotherapy and interferon have been reported [9]. In conclusion, this case indicates the difficulty in diagnosing this rare tumor based only on the clinical-radiological picture, with HRCT differential diagnosis including diffuse interstitial lung diseases, lung cancer with endovascular spreading, vascular tumors, and granulomatous infections. Although a transbronchial biopsy showed a chronic hemorrhagic lung injury, this was only the epiphenomenon either of the radiological pattern or of the interstitial, endovascular infiltrating pathway of the epithelioid vascular tumor cells. Pancytokeratin and common leukocyte antigen (CD45) negativity ruled out either the carcinoma or lymphoma hypotheses, whereas a strong immunoreactivity for vascular markers (CD31 and FLi-1) confirmed the diagnosis of epithelioid hemangioendothelioma. Clinicians should always consider bronchoscopy as a useful and effective tool to better investigate indeterminate and questionable clinical pictures, thereby sparing patients the morbidity and mortality associated with more invasive techniques such as surgical or thoracoscopic biopsy.

## Data Availability

Anonymized data showed are available to editors, reviewers, and readers.

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
