# Peer review of "Watch the Mime Carefully! A Refractory Interstitial Lung Disease"

_diagnostics, 2022, doi:10.3390/diagnostics12071743_

Round 1

Reviewer 1 Report

The clinical case is very interesting. An exceptionally rare disease is described.

Therefore, more precision is desired:

• The clinical case itself is not fully described.

• The diagnostic criteria of  pulmonary hemangioendothelioma  and the compliance of this case with them should be clearly stated.

• The differential diagnosis must be clearly described.

• Discuss the differential diagnosis of histological findings.

• Please describe the follow-up of the case after the biopsy.

Author Response

The clinical case is very interesting. An exceptionally rare disease is described.

Therefore, more precision is desired:

  • The clinical case itself is not fully described.
  • The diagnostic criteria of pulmonary hemangioendothelioma and the compliance of this case with them should be clearly stated.

We are very grateful to Reviewer for her/his comments. Although pulmonary epitheliod hemangioendothelioma is rarely diagnosed by transbronchial biopsy, the chronic hemorrhagic lung injury associated with the interstitial and endovascular pattern of infiltration by neoplastic epithelioid cells immunoreactive for vascular markers suggested the diagnosis of epithelioid hemangoendothelioma.

  • The differential diagnosis must be clearly described.
  • Discuss the differential diagnosis of histological findings.

We are very grateful to Reviewer for her/his kind suggestions.

Radiologically and pathologically, the differential diagnosis of pulmonary epitheliod hemangioendothelioma mainly include metastases, lymphoma, primary lung cancer, with or without endovascular spreading, vascular tumors and infections.

In our reported case, albeit morphological evaluation of atypical epithelioid cells was negatively affected by transbronchial biopsy artifcats, pancytokeratin and common leukocyte antigen (CD45) negativity ruled out either carcinoma or lymphoma hypothesis, whereas strong immunoreactivity for vascular markers (CD31 and FLi-1) consisted with epithelioid hemangioendothelioma.

  • Please describe the follow-up of the case after the biopsy.

The patient was referred to a specialized Oncology Unit but deceased 5 months after the diagnosis.

Reviewer 2 Report

Title   Watch the Mime carefully! A Refractory Interstitial Lung Disease

The manuscript is clear, relevant for the field, However, some points needed to be addressed.

General

Language editing is need

There are multiple grammar errors

Figure 1

There are multiple images. However, the authors did not indicate the value of these multiple images and what did they indicate

The authors described the history of the patient again at the footnote of figure 1, it will be better to replace that with description of each image.

The authors did not give any idea about the basic laboratory investigations of the case

Figure 2

It will be better to add arrow to indicate the main pathology at the images

The authors wrote multiple small paragraph which is not appropriate in scientific writing

The discussion writing is not interesting for reading ...small paragraphs and small sentences

There are many sentences without references

PEH typically occurs as

It is better to avoid starting a paragraph by abbreviation

Re arrange and rewrite the discussion again

Author Response

The manuscript is clear, relevant for the field, However, some points needed to be addressed.

General

Language editing is need

There are multiple grammar errors

Language editing has been performed

Figure 1

There are multiple images. However, the authors did not indicate the value of these multiple images and what did they indicate

The authors described the history of the patient again at the footnote of figure 1, it will be better to replace that with description of each image.

We rearranged images and the text as kindly suggested by the reviewer

The authors did not give any idea about the basic laboratory investigations of the case

Laboratory tests results have been added

Figure 2

It will be better to add arrow to indicate the main pathology at the images

Arrows indicating the main pathology at the images have been added.

The authors wrote multiple small paragraph which is not appropriate in scientific writing

Small paragraphs have been eliminated

The discussion writing is not interesting for reading ...small paragraphs and small sentences

Small paragraphs have been eliminated

There are many sentences without references

References have been rewritten, number 7 in the text was missing

PEH typically occurs as

It is better to avoid starting a paragraph by abbreviation

Abbreviation has been eliminated

Re arrange and rewrite the discussion again

The discussion has been rewritten

Round 2

Reviewer 1 Report

None

Reviewer 2 Report

Accept 

This manuscript is a resubmission of an earlier submission. The following is a list of the peer review reports and author responses from that submission.